

# Enabling smart parking for smart cities using Internet of Things (IoT) and machine learning

Mofadal Alymani[1], Lenah Abdulaziz Almoqhem[2], Dhuha Ahmed Alabdulwahab[2], Abdulrahman Abdullah Alghamdi[2], Hussain Alshahrani[2] and Khalid Raza[3]

[1] Department of Computer and Network Engineering, College of Computing and Information Technology, Shaqra University, Shaqra, Saudi Arabia
[2] Department of Computer Science, College of Computing and Information Technology, Shaqra University, Shaqra, Saudi Arabia
[3] Department of Computer Science, Jamia Millia Islamia, New Delhi, Delhi, India

Corresponding author
Khalid Raza, kraza@jmi.ac.in

## ABSTRACT

With the escalating number of vehicles and the lack of parking spaces, the issue of parking has become a significant problem in major cities as it is a daily occurrence for educational institutions, companies, and government facilities, resulting in fuel wastage and time inefficiencies. In their work lives, employees often face problems when parking their cars in the work parking area. Finding a space for their vehicle can take a lot of time and effort, leading to late arrival for work. On the other hand, security guards have difficulty entering their employees' cars. In this context, our proposed system attempts to address this pressing issue, which consists of two parts: one is a camera at the parking gate that recognizes the license plate using the Automatic Number Plate Recognition (ANPR) algorithm, where the camera captures the license plate and outputs the plate number using the optical character recognition (OCR) technique. After that, the resulting data is cross-referenced with database records for seamless entry authentication. This eliminates the need for security personnel to verify vehicle identities or stickers manually, streamlining access procedures. The second part is a camera in the car parks that distinguishes between vacant and available parking spaces and stores the data collected by the camera in the centralized database, enabling the real-time display of the nearest available parking spots on digital screens at entrance gates, significantly reducing the time and effort spent in locating parking spaces. Through this innovative solution, we aim to enhance urban mobility and alleviate the challenges associated with urban parking congestion, thereby resolving the problem of intelligent parking for smart cities with the help of machine learning.

## INTRODUCTION

A significant increase in the number of four-wheel vehicles, especially private cars, is being observed in the city. As people travel to different locations, these vehicles require parking at various places. Intelligent parking systems, utilizing advanced technologies, are becoming increasingly popular. A key benefit of these systems is their ability to save fuel and reduce

air pollution by minimizing $CO_2$ emissions. By providing real-time information about available parking spaces, smart parking systems alleviate traffic congestion, improve vehicle flow, and enhance parking management. Whether it's a mall, public space, stadium, or hypermarket, there is a growing need for a system that can efficiently detect available parking spots nearby. Such systems must also manage parking locations to prevent misconduct during travel. In line with Saudi Arabia's Vision 2030, the development of smart cities is a priority, with intelligent vehicle parking being a crucial component. These systems help reduce air pollution from cars, a significant concern in urban areas. Various researchers have proposed solutions to address these parking challenges (*Abdelmoamen, 2018*). The system proposed in this study aims to assist in finding the nearest available parking space and securing the parking position to prevent loss. This system also includes number plate detection, which will help identify and store car information. Advanced algorithms for number plate detection can further enhance parking management and be extended to address related issues such as speeding violations, vehicle security, and missing vehicle detection (*Ullah et al., 2024*).

The problem of inefficient parking management is worsening in cities, leading to incorrect and random parking, particularly in government offices, colleges, and institutions. Effective parking management is essential for developing sustainable traffic systems in smart cities. Parking policies play a vital role in ensuring the efficiency of transportation systems and controlling traffic demand. For example, a system utilizing drones to manage air traffic was reported by *Khan et al. (2021a)* as a solution to transportation issues in Saudi Arabian healthcare. Illegal parking can undermine the sustainability of transportation systems (*Aljoufie, 2016*). Finding a parking spot has become a common challenge worldwide, often requiring considerable time and effort. Smart parking technologies, such as cameras or sensors, provide solutions by optimizing parking management and enhancing urban efficiency (*Gandhi et al., 2021*). Smart cities leverage information technology (IT) infrastructure to improve quality of life and promote commercial development. Intelligent parking systems enable quicker parking, reduce the time spent searching for a spot, and offer benefits such as lower pollution, increased safety, real-time parking analytics, reduced driver stress, more efficient use of urban space, and decreased street congestion. (*Lavalle et al., 2020*) On average, it takes five to ten minutes to find a parking space, leading to significant time and fuel loss. It's estimated that over $70 billion in economic value is wasted annually due to the search for parking (*Madakam et al., 2015*). This not only results in lost time and money but also depletes the country's fossil fuel resources (*Arafat, Khairuddin & Paramesran, 2020*). Computer vision, a critical branch of artificial intelligence (AI), plays a significant role in these systems. Much like how human vision informs decisions, AI-powered computer vision enables intelligent systems to perform tasks efficiently based on visual inputs (*Awalgaonkar, Bartakke & Chaugule, 2021*).

## Optical character reader

Determination of distances, distinguishing objects, identification of time for reaching a place, and many more such decisions can be made by the human mind based on the site. A similar kind of technique can be implemented with the help of computer vision for

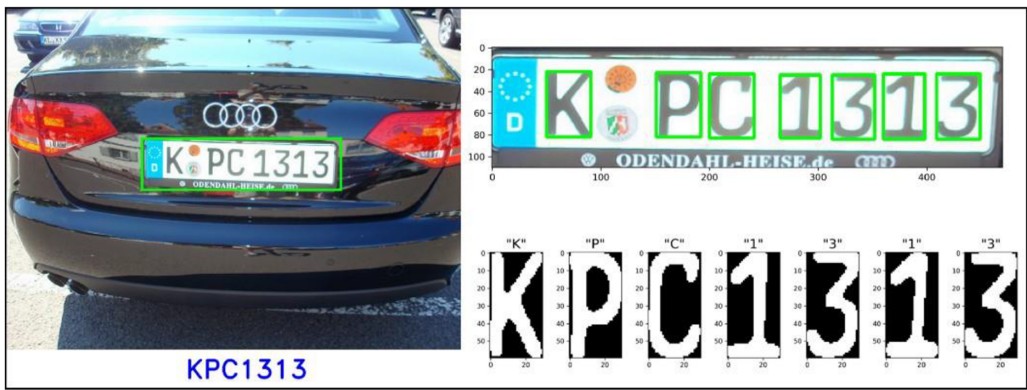

**Figure 1** Detect and recognize vehicle number plates (*Weihong & Jiaoyang, 2020*).

machines and mechanical devices. AI empowers computers to make decisions based on what is observed with the help of computer vision (*Awalgaonkar, Bartakke & Chaugule, 2021*). The human brain and the optical nervous system, including the retina, computer vision, trains, robots, or computational devices, perform all the activities done by humans site for calculating time and distances, including object identification (*Zou et al., 2020*). Once the system is given proper guidelines and training to identify some objects and make respective decisions, AI helps make meaningful decisions. These systems are empowered by various algorithms that run on the data collected by computer vision (*Rademeyer, Barnard & Booysen, 2020*). Once the data reaches the system and proper algorithms act upon it, it can perform swiftly and analyze many input sets to solve various problems. The advent of optical character recognition (OCR) in 1974 marked a significant milestone in computer vision and pattern recognition. Developed initially to interpret printed text in images automatically, OCR has since evolved into a versatile technology with applications spanning various industries and domains. OCR employs complex algorithms to detect and extract text from images or scanned documents. These algorithms analyze individual characters' shapes, patterns, and spatial relationships to recognize and interpret them accurately. Over the years, advancements in image processing techniques, machine learning algorithms, and computing power have greatly improved the accuracy and efficiency of OCR systems. One of the primary applications of OCR is in document processing and management, where organizations rely on OCR to digitize vast amounts of printed documents and convert them into searchable and editable digital formats. In addition, OCR plays a crucial role in automating data extraction and entry processes. In finance, healthcare, and logistics, OCR-powered systems capture critical information from documents and input it directly into databases or enterprise systems. This automation reduces manual labour and human error, accelerates decision-making, and improves operational efficiency. OCR also plays a crucial role in number plate recognition systems by enabling automated extraction and interpretation of alphanumeric characters from vehicle license plates captured in images or video streams (Fig. 1) (*Huang & Liu, 2019*).

## Automatic number plate recognition

Automatic Number Plate Recognition (ANPR) (*Qadri & Asif, 2009*), also known as Automatic License Plate Recognition (ALPR), is a technology that uses OCR algorithms, image processing, and pattern recognition techniques to automatically detect, read, and interpret license plate numbers from images or video streams captured by cameras. Usually, the collected data is subsequently cross-referenced by applications or systems to take meaningful results and provide decisions. The two primary components of this technique are plate detection and plate recognition (*Hendryli & Herwindiati, 2020*). The goal of plate detection is to locate the plate inside the entire camera frame. A plate segment is forwarded to the second stage when one is found in an image, plate recognition, which deciphers the alphanumeric characters on the plate using an OCR algorithm. The ANPR systems typically rely on cameras strategically positioned to capture images of vehicles and their license plates. These cameras may be stationary (such as those installed on highways, toll booths, or parking lots) or mobile (mounted on vehicles or handheld devices used by law enforcement). Before analyzing the captured images, preprocessing techniques are applied to enhance image quality and improve the accuracy of license plate detection and recognition. This may involve noise reduction, contrast adjustment, image resizing, and normalising lighting conditions. One of the primary tasks of an ANPR system is to locate the license plate within the captured image. This is accomplished through image processing algorithms that identify regions of interest based on colour, shape, texture, and aspect ratio characteristics. Once potential plate regions are detected, further analysis is performed to refine and isolate the license plate area. After localizing the license plate, the next step is to segment individual characters or symbols comprising the license plate number. This involves separating the characters from each other and any surrounding noise or artefacts in the image. After the segmentation of characters, OCR algorithms are applied to recognize and interpret the alphanumeric symbols. These algorithms analyze the visual features of each character and compare them against predefined templates or statistical models learned from training data. *Khan et al. (2021b)* provided an application-based system for working in the academia domain, which facilitates users to avoid any congestion in the premises where several vehicles and services are provided. *Alshahrani et al. (2022)* suggested several techniques to prevent the problems associated with cyber threats to personal data and information. The recognized license plate numbers are typically used to query databases containing vehicle registration information, owner details, and relevant records. Based on the query results, ANPR systems can trigger various actions or alerts, such as logging vehicle movements, enforcing traffic regulations, managing parking facilities, or identifying vehicles of interest (*Du et al., 2012*).

## REVIEW OF LITERATURE

In all cities, there is a fundamental problem with parking. Finding parking space takes more time for people. Some people make mistakes and park in strange places (*Carrera García, Recas Piorno & Guijarro Mata-García, 2022*). Employees, for instance, may have trouble finding a parking spot for their cars, which causes them to be late for work. Parking lots

frequently suffer from poor management, coordination, and planning without technology (*McCoy, 2017*). Dedicated parking management solutions streamline the entire process to ensure speed and accuracy while reducing the need for human interaction in regular tasks (*Ahmed, 2019*). These advantages span parking lot owners, governing bodies, final consumers, and the entire supply chain. The average driver spends 17 h per year looking for parking on streets, in lots, or garages. Not only do vehicles struggle with this, but also nearby shops and companies (*Agarwal & Bansal, 2024*). The Kingdom of Saudi Arabia has created a vision for 2030 to help the nation forge a progressive future. Saudi Vision 2030 is a strategy plan to diversify Saudi Arabia's economy, lessen its reliance on oil, and expand public service areas like health, education, infrastructure, leisure, and tourism. This study is entirely dependent on the use of AI to foster digitization and improve the standard of living for the residents and citizens of Saudi Arabia. The proposed research will benefit Saudi Arabia in managing Smart City activities related to traffic management and vehicle parking issues. The study in this article focuses on developing a digital system that artificial intelligence powers to achieve the nation's Vision 2030. To enhance road safety inside residential and institutional compounds, an automated speed radar system was proposed by *Al-Hasan et al. (2022)* that comprises three central units - a camera unit, a speed radar unit, and an ANPR unit. The system was tested with 3,200 high-definition number plates from Qatar University security gate. The authors in *Koumetio Tekouabou et al. (2022)* presented an IoT and set-based regression model for predicting parking space with regression analysis. *Abbas et al. (2023)* reported the SCOPE model for identifying parking lot statuses using YOLOv3 and replacing the AlexNet. The model depicted by *Harish Padmanaban & Sharma (2024)* optimizes identifying and utilising open parking spaces in traffic flow management and road mapping using sensors and cameras.

As per the study given by *Anekar, Yeginwar & Sonune (2022)*, the automatic gate control mechanism integrated with the vehicle license plate can be used to recognise the number present in the database of the parking zone. The main objective was to find a system capable of identifying the correct vehicle entering a parking space and opening its gates automatically for improved efficiency and convenience at all access points. The automatic gate control system will be able to operate without human assistance. It will also be able to identify license plates from approaching vehicles and decide whether to allow them in. The system, built around a PIC micro-controller and a standard PC with a video camera, captures and interprets video frames with visible vehicle license plates. The recognition rate of the suggested system, which was constructed using MATLAB, Proteus, and Micro C, is 98 percent (*Ahmed, 2019*). *Aditya et al. (2023)* proposed an IoT and cloud-based Intelligent Parking System that collects real-time data, sends it to the cloud, and recommends the driver a suitable nearby parking space for the vehicle. The system uses Raspberry Pi, NodeMCU, radio frequency identification (RFID), and infrared (IR) sensors. However, the accuracy of this system has not been assessed. A hybrid approach consisting of a support vector machine (SVM) and artificial neural networks (ANNs) has been proposed by *Ali & Khan (2023)* to enhance the security and privacy of the automated parking allocation system. *Khan et al. (2019)* suggested several mechanisms in the security of applications that reside in local or cloud-based services. This approach considers credibility, availability, and

honesty as critical parameters. They reported an accuracy of 96.43 percent in predicting and eliminating malicious or compromised nodes. *Awaisi et al. (2023)* propose an Industrial IoT (IIoT) enabled smart parking system based on a deep reinforcement learning framework that comprises smart cameras, fog nodes, and a cloud server. Deep reinforcement learning, such as a deep Q-learning algorithm, is deployed on fog devices for vehicle classification and intelligently allocating vacant parking slots. The reported accuracy of parking slot image detection was between 84–99 percent for the number of parking slot images from 200 to 2,000. A long-range autonomous valet parking framework proposed by *Khalid et al. (2023)* aims to optimize the path planning of autonomous vehicles to minimize distance while ensuring a high-quality user experience. The framework involves autonomous vehicles picking up and dropping off users at designated spots and autonomously navigating to parking areas. They introduced a Double-Layer Ant Colony Optimization (DLACO) algorithm for iterative optimization. A deep Q-learning network (DQN) algorithm is also presented to help with quick decision-making in dynamic environments. Experimental results demonstrate the effectiveness of both DLACO and DQN-based approaches in achieving significant performance improvements.

*Henry, Ahn & Lee (2020)* provided a systematic review of all the available technologies for real-time testing and simulation of the algorithms that will be used to integrate computer vision. The integrated and mentioned technologies in this study used the ANPR systems. Several researchers like *Alshahrani & Khan (2023)* reported some exciting mobile-based applications that can be used for user analysis. Similar applications can be designed in terms of smart parking, which can understand user behaviours. These applications can be beneficial for parking suggestions as well. Recognition algorithms allow ANPR technology to detect and identify vehicles based on their number plates (*Mufti & Shah, 2021*). A successful ANPR system deployment may need additional hardware to increase accuracy, even with the most significant algorithms. By identifying pertinent earlier work, examining and presenting a review of extraction, segmentation, and recognition techniques, and providing suggestions on future trends in this field, this research intends to advance the state of knowledge in its ANPR built on Computer Vision algorithms. The study proposed by *Anekar, Yeginwar & Sonune (2022)* explains a system that will be capable enough to identify the plot space for the parking and will be able to track the occupancy in real-time for the cars or the vehicles available in that zone. The algorithm proposed in this study uses the zenith photographs using artificial vision (*Abdelmoamen, 2018*). By filtering, thresholding, extracting the contour, and approximating the parking spots of an empty parking lot to a polygon, the described system initially detects the available parking spaces semi-automatically. The system could identify the presence of a vehicle in the space, which provided 98.21 percent results. The conventional neural network used for this algorithm determines the features' size and location to identify the spaces mapped to a vehicle.

Automated video surveillance systems are vital for ANPR and crowd safety, yet deep learning algorithms encounter hurdles like prolonged training and data scarcity. Deep transfer learning and domain adaptation offer promising solutions by simplifying training, enhancing model generalization, and addressing data scarcity. *Himeur et al. (2023)* presents a comprehensive review of these techniques in video surveillance, outlining their

advantages, challenges, and future directions. To optimize parking space utilization, an IoT-based model was introduced by *Rajyalakshmi & Lakshmanna (2024)* that employs a Hybrid Deep DenseNet Optimization (HDDNO) algorithm, combining machine learning and deep learning techniques. *Jabbar, Tiew & Shah (2024)* presents another IoT-driven Smart Parking Management System utilizing Long Range Wide Area Network (LoRaWAN) technology to overcome traditional communication and cost limitations. The system employs intelligent sensing nodes with Arduino UNO microcontrollers and sensors to detect vehicle occupancy, transmitting data *via* LoRaWAN. It provides real-time parking updates through an accessible graphical interface and operates independently on solar power. However, the system has various limitations, including its reliance on a single gateway for data transmission, which can not be scaled efficiently to cover larger parking areas or dense urban environments where multiple gateways might be necessary to ensure adequate coverage and robust connectivity. Further, using non-switch-type magnetometers may pose challenges in accurately tracking parking durations, potentially leading to inaccuracies in occupancy data and subsequent billing calculations.

Several authors have been working on the recognition of number plate systems. *Rakshe & Dongre (2024)* presented a model using the convolutional neural network associated with optical character recognition. This model uses edge detection with the help of Hough's and Sobel's methods. Yet another model was proposed by *Varma P et al. (2020)*, which emphasized using a convolutional neural network and optical character recognition without any edge detection technique. The prescribed model uses the MobileNet framework to reduce images related to the number plates. *Angelika Mulia, Safitri & Gede Putra Kusuma Negara (2024)* proposed a faster R-CNN method using Tesseract-OCR libraries in Python programming language to detect the number from the plate. No edge detection was required in this method, and the authors used YOLOv8 for the plate detection. Deep learning was integrated with the EasyOCR and MobileNetV2 frameworks by *Devisurya et al. (2024)*. The authors also presented several other models in which traditional methods were used, comprising artificial neural network typical number plate detection techniques. The model proposed in this study uses the KNN technique to identify and classify the numbers from the images. The model proposed in this study uses the KNN technique to identify and classify the numbers from the pictures. The proposed model emphasizes using YOLOv8 and SVM to provide a better alternator for detecting the numbers. Using the bounded box technique, the canny edge detection technique is applied to identify number plates. This makes it more powerful and effective to ensure the identification of a number plate when the car enters the parking station. The model proposed is solely represented for parking locations, and therefore, it is a necessary condition to identify the number plate correctly. It is assumed that light conditions and atmospheric issues are less in parking locations, making the model more potent with the help of modern cutting-edge techniques for identifying numbers from the car plates.

Table 1 reflects a comparative study of the proposed model with several well-known state-of-the-art techniques in the domain. It also represents the technical aspect of the methods used in previous models compared to the present. The comparison with the state-of-the-art model in Table 1 illustrates various techniques for detecting number

**Table 1 Comparison of the proposed model with other state-of-the-art methods in terms of techniques, OCR, edge detection.**

| Citation | Technique | OCR | Edge detection | Model used |
|---|---|---|---|---|
| *Rakshe & Dongre (2024)* | CNN | Yes | Hough's, Sobel's | imageNet |
| *Varma P et al. (2020)* | CNN | Yes | No | MobileNet |
| *Angelika Mulia, Safitri & Gede Putra Kusuma Negara (2024)* | Faster R-CNN | Tesseract-OCR | No | YOLOv8 |
| *Devisurya et al. (2024)* | Deep Learning | EasyOCR | No | MobileNetV2 |
| *Gopikrishnan et al. (2023)* | Haar Cascade | EasyOCR | No | CV2 |
| *Purnomo & Maharani (2023)* | LDR | ANPR | No | Application |
| *Elfaki et al. (2023)* | CNN | EasyOCR | No | MobileNetV2 |
| *Narendra et al. (2023)* | CNN | OCR | No | Application |
| *Srivastava & Goel (2023)* | Deep Learning | OCR | No | Yolo |
| ANPR (proposed model) | KNN | Tesseract-OCR | Canny | YOLOv8,SVM |

plates. Several models were described by multiple researchers in this field. Most of the models make use of CNN architecture. All the models use OCR techniques to identify the number plate characters. The model presented in this study uses an edge detection technique to ensure that the number plate is recognized efficiently. Different authors used several image processing models, and the prominent models were MobileNetV2 and Yolo. The proposed study ensures that the combination of techniques utilizing the prediction of the number plates ensures a high probability of identifying the numbers with deep accuracy. This study's scope includes all locations requiring parking spaces, including businesses, government agencies, and educational institutions throughout the Kingdom of Saudi Arabia, ensuring these locations' safety and security. As per Vision 2030 of the Kingdom of Saudi Arabia, the digitization of various processes is expected to be expanded to enhance the lifestyle of the citizens and residents. The work proposed in this study will empower the digitization of parking systems, saving the country's wealth. The integration of Hindi-Arabic Numerals is needed for the number plate recognition in the Kingdom. The model proposed is an enhanced version compared to other state-of-the-art techniques to ensure that the detection of plates is done despite the Arabic language digits and characters.

The contributions of this work are as follows:

- Development of an intelligent parking system leveraging the ANPR algorithm integrated with OCR for automated vehicle authentication and seamless entry.
- Implementation of a camera-based system for real-time identification of vacant parking spots, enabling efficient parking management.
- Integration of centralized database records to streamline access procedures and reduce dependency on manual security checks.
- Introduction of real-time digital displays to guide drivers to available parking spaces, reducing time and fuel consumption.

- Advancement in smart city technology by addressing urban mobility challenges and optimizing parking efficiency using machine learning techniques.

# MATERIALS AND METHODS

The study proposed in this article focuses mainly on developing a model required by several Smart cities to ensure proper parking spaces in populated areas. Usually, the procedures specified in these studies by several authors implement systems across moving roads, toll taxes, closed circuit televisions, *etc.* However, this study uses convolutional neural networks along with KNN classification. The novelty of the research related to this article is that KNN with OCR techniques is used to identify the number plates. The complete process used in this study is summarized as follows:

## Working of the proposed system

The administrator will initially record the security guard's data to access the system. The security guard will record car owners' data, including license plate numbers, in the database. Then, if the vehicle owner wants to park it in the parking, the gate camera will scan the license plate number with the ANPR algorithm, which recognizes the vehicle and extracts the license plate number with OCR. The license plate number is stored and compared against the database. The gate will not open if the plate number isn't in the database. The gate will open, and the nearest parking space's number will be displayed on a digital screen if the database contains the license plate number. Only authorized vehicle owners can enter the parking lots to ensure safety. There will be a camera in the parking space to detect vacant and occupied spaces, and the data will be sent to the database to display the nearest car park number on the digital screen. To save time, instead of searching the car parks for a vacant park (Fig. 2). When the car arrives, the camera automatically captures the image, and the plate is detected and recognized. After that, the system searches for the plate number in the database; if it exists, the gate is opened, and the nearest parking number is displayed after searching for it in the parking space database. Otherwise, the gate does not open. The Algorithmic workflow is depicted in Fig. 3.

## Image pre-processing: edge detection, segmentation, plate localization, number plate extraction and OCR

Figure 4 depicts the pre-processing steps; first, the original image is converted to a grey image, and then we convert it to a canny image format. The next step is to identify the contour and make sure that it is a rectangle or square to determine the location of the plate. In the initial level, the captured images are passed into the processing phase. The edges of the number plate are detected. In the next step, the system creates a CSV file to register the car owner's information, including the plate number; the system will read the CSV file and compare it with the plate number extracted from the image. If the plate number is found, the system registers the number in another file called history from the database.

### Color conversion

Conversion of the input image to gray-scale and enhancement of contrast using an adaptive threshold. The RGB captured images are converted to gray-scale for proper interpolation

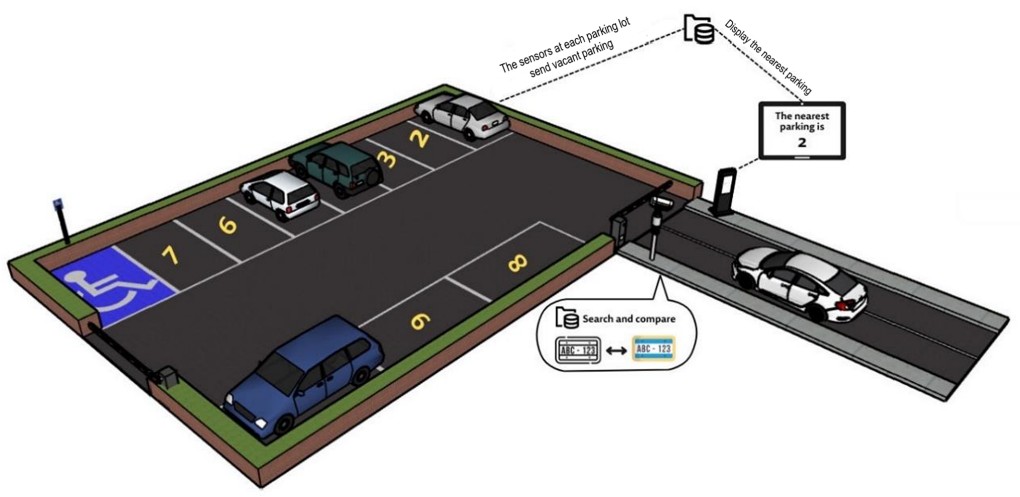

**Figure 2**  Demonstration of the parking scenario.

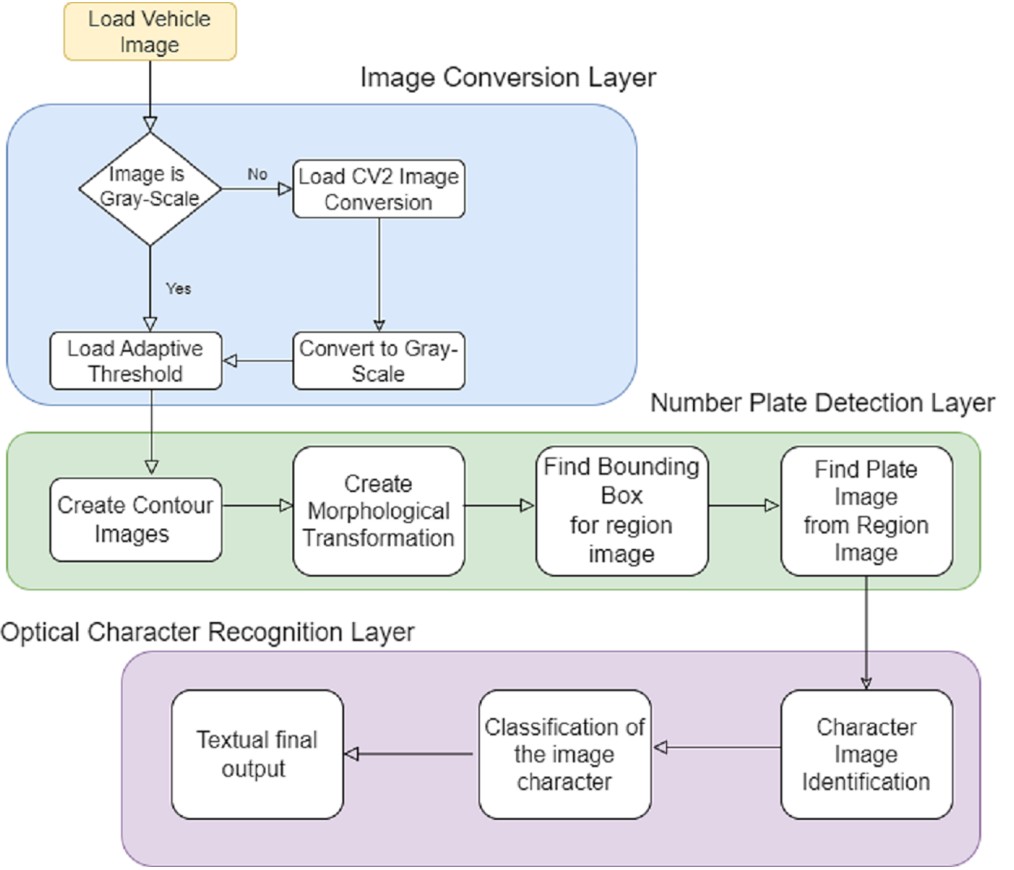

**Figure 3**  Algorithmic workflow of the proposed system.

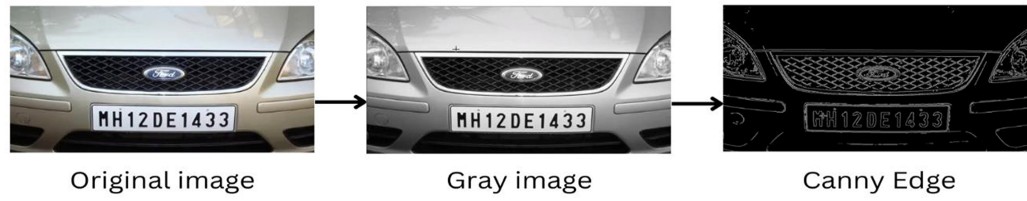

**Figure 4   Image conversion for number plate detection.**

of the results with the help of steps given in Algorithm 1. To enhance the accuracy of the model the image captured is processed using several techniques. The images are segmented using the proper features and classifiable objects.

### Edge detection and segmentation

We used the Canny edge detection algorithm (*Canny, 1986*) to detect the boundaries of objects in the image. The Canny edge detection is used in this model to avoid the noise that might appear in the gray-scale image. The initial stage for Canny edge detection results in the removal of the noise using the Gaussian filter (*Abdellatif et al., 2023*). *Manana, Tu & Owolawi (2021)* represented a technique for edge detection by matching the extracted vehicle license plate with the plates detected earlier. In our case, we have used the nearest Euclidean distance equation for the identification of the various characters in the model.

### Contour detection and plate localization

We used contour detection to identify potential regions containing the number plate based on shape and size constraints, especially in Hindu-Arabic numerals (*Elfaki et al., 2023*).

### Number plate extraction

Reading and cropping the number plate from the image using the $x_{min}$, $x_{max}$, $y_{min}$, and $y_{max}$ coordinates from the number plate. Once the number plates are extracted we tend to provide them to the segmentation stage to enhance the classification of the plate.

### Reading characters using OCR

At the end of the complete pre-processing steps, the suggested study applies an OCR algorithm like EasyOCR and Tesseract to read characters on the number plate. In our case, both methods of reading characters are adapted to ensure that the Hindu-Arabic numerals and digits are captured. Once the complete set of characters or digits is identified in the buffer stack, it stores the number and characters recognized in the database. Several researchers have proposed different models that use various OCR techniques. However, in this case, Tesseract OCR helps us determine the number of plates with the help of bounding boxes and edge detection. The models, compared to state-of-the-art stables, use different edge detection techniques. However, we implement the use of canny edge detection in this situation.

---

**Algorithm 1** Image Pre-processing and detection of number plate

---

Input (Image)

Gray_Image = cv2.cvtColor(Image,cv2.RGB2HSV.values)

#Use of OpenCV2 for creating the Gray Scaling

Foundation_Image = cv2.adaptiveThreshold(Gray_Image)

List = ContourImages(Foundation_Image)

**for all** values in List **do**

    Value = morphological_transform(List)

**end for**

Get_char = image_Segment(List)

**for all** Get_char nch in the List **do**

    Box_Finding = findBoundingBox(nch)

    **if** Box_Finding == TRUE **then**

        Image = find_Plate(Image)

    **else**

        Return Image

    **end if**

**end for**

---

## Number plate detection algorithm

Figure 4 displays the localization and detection of the number plates in our model that requires processing the colour images by converting them to greyscale canny images. The de-saturation for the colours represented inside the scale of the number plate is split into the core colours, thereby measuring the pixel values (*Sood, Nandakumar & Rajkumar, 2021*). OpenCV2 makes it possible to split assorted colours into respective RGB values with the help of the following equation:

$$E = 0.1(3 * R(i,j) + 6 * G(i,j) + B(i,j)) \tag{1}$$

where S(i, j) represents the spectrum of the image, and the R(i, j), G(i, j), and B(i, j) represent the red, green, and blue colour values splitter for the identification of the image. We used the adaptive threshold method to accurately identify the proper image alphabets and texts. This method works better for variable light conditions during the input images, both during the day and at night. This method calculates the threshold value for all the regions in the image that are responsible for providing any variable elimination. Cv2.adaptiveThreshold function is used to capture the value of all the threshold parameters.

## Machine learning models used for training and testing

In this work, we have utilized SVM and YOLOv8 (*Jocher, Chaurasia & Qiu, 2023*) models for training and testing the model. SVM is a supervised learning algorithm primarily used for classification tasks which works by finding the optimal hyperplane that separates data points into different classes. In a high-dimensional space, SVM identifies the margin between classes, aiming to maximize the distance between the nearest data points (support vectors) of each class and the hyperplane. This makes SVM effective in handling both

linear and non-linear classification problems, with kernel functions allowing it to model complex relationships between features. YOLOv8 (You Only Look Once, version 8) is a state-of-the-art object detection model that excels in real-time detection tasks. Unlike traditional detection algorithms that apply sliding windows over the image, YOLOv8 processes the entire image in one go, predicting bounding boxes and class probabilities simultaneously. This single-stage approach makes YOLO highly efficient. YOLOv8 incorporates improvements in accuracy and speed over its predecessors by leveraging better feature extraction techniques and neural network architectures, making it well-suited for applications requiring fast and accurate object detection, such as license plate recognition.

## Datasets used

The complete experiment was conducted on 788 images of English number plates, out of which 355 images were taken from the open Roboflow dataset (*Mehako, 2023*) and 433 images from the Kaggle (*Larxel, 2020*). Also, 593 images of Arabic number plates were considered from *Ammar et al. (2023)*. The dataset used in this study contains several fields for edge detection in a vehicle's number plate. The dataset contains several fields. The most promising fields needed in this study are:

- Image name: Contains the name of the image for the vehicle
- Image of vehicle: Contains the Image for the vehicle number plate
- Width: Contains the width of the number plate for the vehicle
- Height: Contains the height of the number plate for the vehicle
- Depth: Contains the depth of the number plate for the vehicle
- xmin: Contains the x: minimum coordinate value for the bounded rectangle box of the number plate for the vehicle
- ymin: Contains the y: minimum coordinate value for the bounded rectangle box of the number plate for the vehicle
- xmax: Contains the x: maximum coordinate value for the bounded rectangle box of the number plate for the vehicle
- ymax: Contains the y: maximum coordinate value for the bounded rectangle box of the number plate for the vehicle

During the testing of the model, the augmentation is done with several images from the general search in a web-based search. We thought the lighting conditions should be more precise due to the limitation of the camera resolutions. This module identifies the final image as the output, extracted from the grayscale image created in the previous step. To get the nearest value of the character present in the image, the similarity equation used is as follows:

$$S(x1, y2) = 1 - d(x1, y2) \tag{2}$$

where d (x1, y2) is the Euclidean distance value between two character position vectors in the greyscale image. The calculation of d is done with the equation:

$$d(x, y) = \sqrt{\sum_{i=1}^{n} (x_i - y_i)^2}. \tag{3}$$

## Implementation of the proposed system

Python's versatility, open-source nature, and extensive libraries make it ideal for integrating machine learning modules such as SVM and YOLO8. This work uses OpenCV for real-time license plate detection, TensorFlow for data automation and model tracking, and EasyOCR (based on PyTorch) for efficient text recognition. The system is programmed using Visual Studio Code, where OpenCV captures images, converts them to grayscale, applies edge detection, and identifies contours to locate the license plate. The text is then extracted with Python-Tesseract, while EasyOCR handles the recognition of Hindu-Arabic characters common in the Saudi Arabian region. The extracted plate numbers are compared with a CSV file of registered car owners, and if matched, the event is logged. An Arduino UNO, connected *via* PyFirmata, controls the parking gate using a servo motor (*Steiner, 2009*). IR sensors track parking availability, with the Arduino IDE managing sensor inputs to update a digital display on available spots. SQLite stores vehicle license plates and check-in times, while Google Colab is used for developing machine learning models in a Jupyter Notebook environment, providing access to necessary computational resources.

Figure 5 represents the complete electronic circuit by fixing to ensure all parts work. Using Arduino IDE, we wrote the code to defend the IR sensor at each parking spot that senses the parking status and uploaded it on Arduino UNO. The sensors send continuous signals to the computational unit which determines what is displayed on the screen so that the LCD will show the nearest parking according to the readings. The digital screen will show the entire parking lot if there are no parking lots. The conditional If-Else Looping is done to ensure that all the parking locations and spots are available. The design prototype of the proposed model is depicted in Fig. 6.

## RESULTS

The experimental evaluation of the model presented in this study was done to identify the classification accuracy and performance of the model. The instrumentation for the entire study was set up with the help of a demonstrator prototype model presented in section 3. The experiment is conducted with the help of image data sets to identify the number plate recognition. The complete experiment is done on 788 images of English and 593 images of Arabic number plates, as discussed in 'Number plate detection algorithm'. The proposed model was trained on 80% of the dataset and tested on the remaining 20%. These images were loaded within the computational environment created on an Intel core i5 machine having 4.5 GHz frequency with 16GB RAM. The epoch size was considered 100 to ensure proper identification of the image processing with the help of the computational environment. The CPU utilization and the execution of the data set with two models, SVM and YOLOv8, are carried out to test the proposed architecture. The value of 100 epochs is chosen due to the limited size of the data set. The expected results were excellent compared to the previously defined state-of-the-art models. All the data sets were executed in the Python environment with the help of the Ultralytics package *Jocher, Chaurasia & Qiu (2023)*. The trained model was tested with the help of image files and the demonstrator

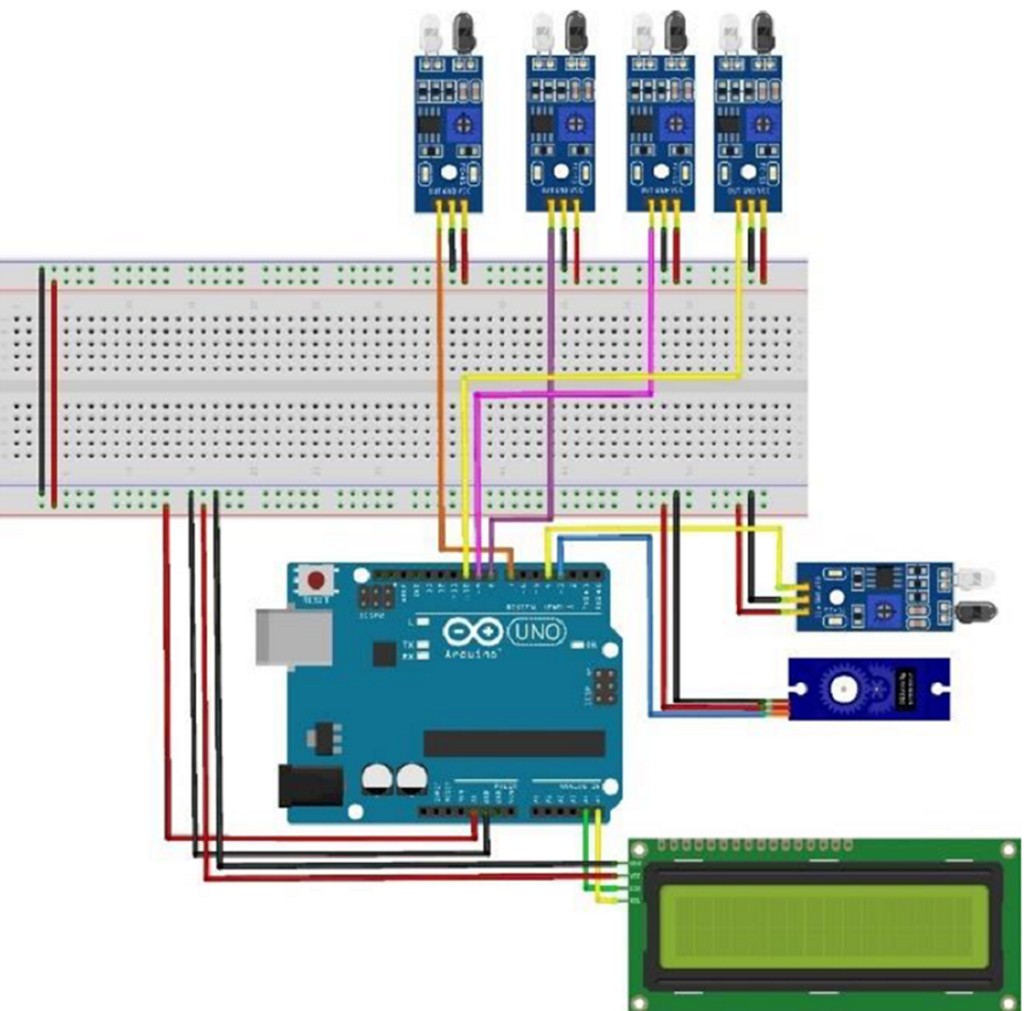

**Figure 5** Complete circuit diagram of the proposed system.

video to track the number plate. An approximate speed of 0.3 ms for the preprocessing, 2.9 ms for the interference check, and 3.0 ms for the postprocessing of an image was averaged for the complete data set. The Arduino UNO 3 model was used in connection with the IR sensors. All the infrared sensors are regular, with a maximum approachable range of 5 m.

## Outcome of the proposed model

Contextual data determines the detection of a number on our number plate. Therefore, the model comprises the dataset with images in the natural format during natural daylight. They are not processed images, so the model's accuracy will be higher than expected. The images sampled and prototyping done with the help of our model performed much better with variable distances of the car number plate under several hazy, windy, and clear weather conditions. However, it must be revealed that the proposed model only detects the number plates used in the parking zones. Usually, the parking zone detection panels are presented in adequately lit areas without extreme weather conditions. Despite the scenario,

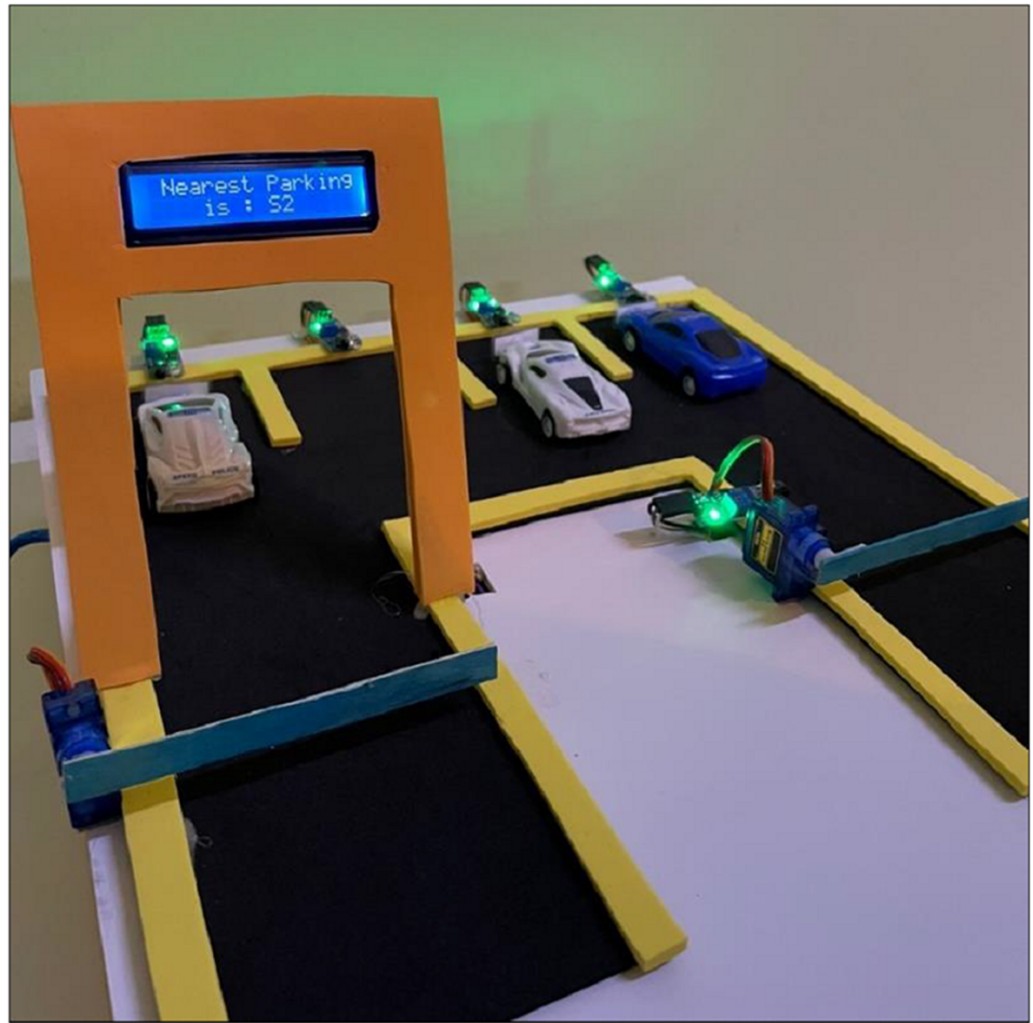

**Figure 6** **Design prototype for the proposed model.**

the model is tested in various situations. The elimination conditions are variable according to daytime and night-time hours. The model was tested under all these circumstances to find all possible mismatch errors. Various classification accuracy rates for the model represented in this study were identified with variable dataset sizes.

We have utilized SVM and YOLOv8 models in our work and compared their performances for the detection. The model achieved several consistent accuracy percent levels ranging from 94.6 to 99.8 for the recognition of the number plate under these circumstances for varying numbers of plates, as shown in Fig. 7. The overall accuracy for identifying the number plates comprises plate detection and character segmentation, followed by the recognition of the complete number. The accuracy of finding all these factors was achieved at a consistent rate of 96.9 percent. We observed that the detection accuracy of YOLOv8 is slightly better than the SVM model. Figure 8 presents a comparison

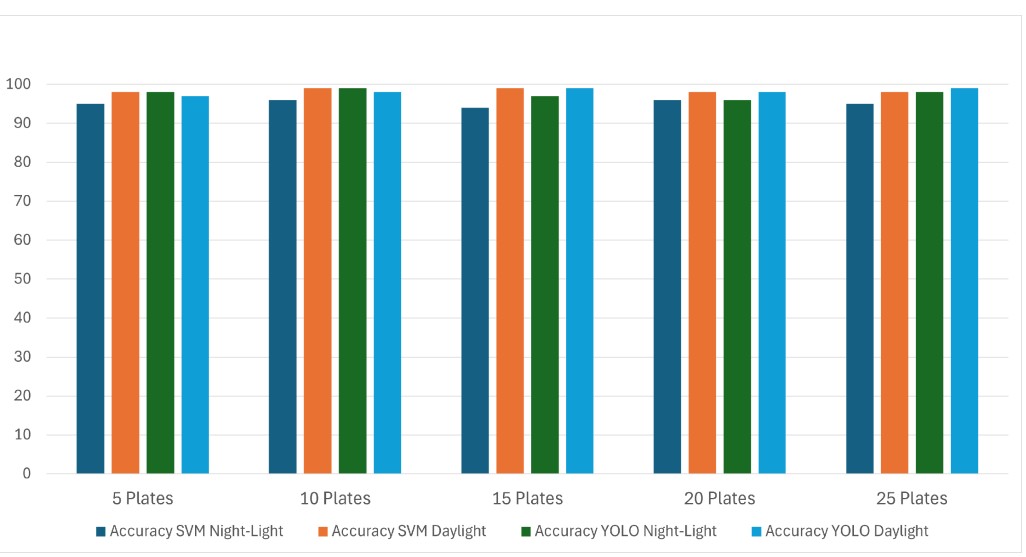

**Figure 7** **Accuracy for character recognition in variant light.**

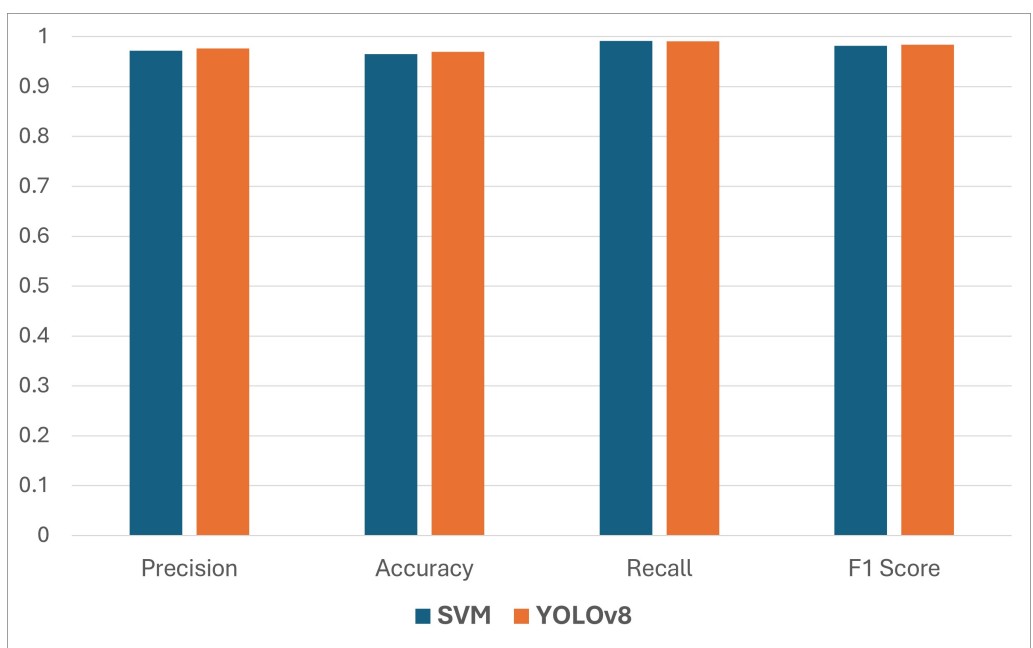

**Figure 8** **Classification performance of the SVM and YOLOv8 models for the proposed system.**

of the classification performance between the SVM and YOLOv8 models for the proposed system, evaluated using precision, accuracy, recall, and F1 score.

We have also tested the model several times under variant light conditions and the nature of the number plates (dirty, clean) for several iterations. Figure 9 presents experimental results showing the performance of the proposed model under three different conditions:

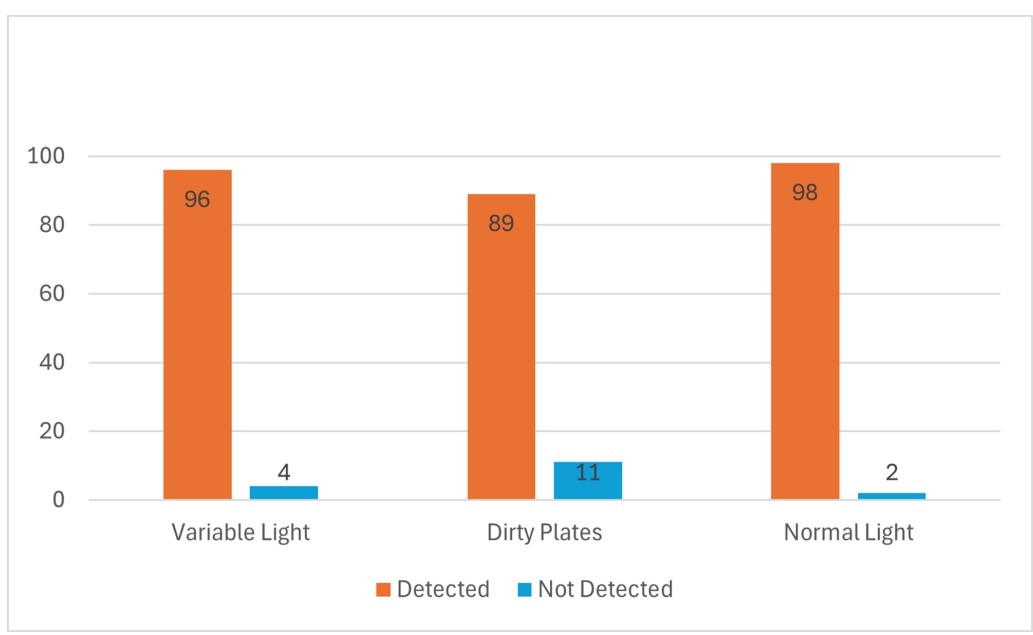

**Figure 9** **Results showing the performance of the proposed model for 100 iterations under three different conditions: variable light, dirty plates, and normal light.** The results are categorized as either Detected or Not Detected.

variable light, dirty plates, and normal light. The results are categorized as either "Detected" or "Not Detected", indicating the success or failure of the model in identifying license plates under these conditions. The model demonstrates high detection accuracy across all three conditions. However, its performance decreases slightly when faced with dirty plates, likely due to occlusions or reduced clarity in the visual input. In normal lighting conditions, the model performs optimally, with very few missed detections. Overall, the model is effective in various scenarios, but conditions like dirt on plates or significant lighting variations can introduce minor detection challenges.

We compared the performance of our proposed model with several existing methods, as shown in Table 2. In our study, we specifically focused on comparing our results with existing number plate recognition methods documented in the literature. To ensure a relevant and meaningful comparison, we limited our analysis to methods that handle English and Arabic number plates. This focus is due to the extensive research and established methodologies available for these scripts, which provide a robust baseline for evaluating our approach. Unlike the other approaches that only support either English or Arabic number plates, our model is capable of accurately detecting both English and Arabic, leading to significantly improved results. Further, our model consistently achieves higher detection rates across various conditions, including variable lighting and dirty plates. The superior results of the proposed model can be attributed to its optimized architecture, which efficiently handles challenging conditions while maintaining better accuracy.

The prescribed model in this study, *i.e.,* YOLOv8 (*Jocher, Chaurasia & Qiu, 2023*), has an overall processing time to identify the characters available in the license plates for the testing

**Table 2 Comparing the performance of the proposed model with other state-of-the-art methods.** All the results reported here were taken from the original works.

| Methods | Language support | Accuracy (%) |
|---|---|---|
| *Saleem et al. (2016)* | English only | 84.8 |
| *Kabiraj et al. (2023)* | English only | 85.0 |
| *Elfaki et al. (2023)* | English only | 89.0 |
| *Angelika Mulia, Safitri & Gede Putra Kusuma Negara (2024)* | English only | 90.0 |
| *Islam, Sharif & Biswas (2015)* | English only | 92.0 |
| *Khan et al. (2022)* | English and Arabic | 92.8 |
| *Abdellatif et al. (2023)* | Arabic only | 93.0 |
| *Safran, Alajmi & Alfarhood (2024)* | English letters and Arabic digits only | 96.1 |
| *Jawale et al. (2023)* | English only | 98.5 |
| Our proposed model | English and Arabic | 96.9 |

dataset collected from open-source data. The model has an overall CPU performance time even after including the Arduino functionality and the website input–output characteristics. The application of Arduino and Web-based data entry contributes to a tiny fraction of the time needed for the system's overall processing as per a study given by *Khan (2021)*. Thereby making the system one of a kind. The model proposed is a prototype for parking zones where the weather conditions are not extreme. 'Data Privacy and Integrity' represents various conditions under which the system is likely to fail. These conditions should be considered while testing the system proposed in this model. The prediction of the number of plates with ambient light conditions and the proper arrangement of the camera are not enough for the model. The availability of light, adequate location of the camera, organized and systematic number plate, and written letters and characters are the essential factors to be considered. The images tested from the open-source data sets and hypothetical data sets were recognized with a higher accuracy rate, which means the eligibility of the acceptance model is high. We have tested the model in a luminous place; however, it is expected that the model will yield anonymous results under weak light conditions.

Multiple authors and researchers have applied the automatic number plate recognition system to various domains of traffic security and safety. Machine learning models invoke the accuracy and completeness of the algorithms used for multiple vehicle safety and traffic management domains. The system developed under this research identifies the number in combination with IoT devices to provide a complete solution for finding the nearest parking in large spaces such as government buildings, malls, airports, public parking, and densely populated sections for keeping the cars. The system proposed in this structure works in three different models. During the first phase, the number plate detection occurs concerning the ANPR algorithm. The boundaries of the number plate are recognized, and then the images are extracted to a grayscale. The number plate's alphanumeric characters are extracted from this grayscale model image. We use the K-nearest neighbor model (KNN) (*Zhang, 2016*) to detect the number on the plate. "The kNN classifier is used to classify unlabeled observations by assigning them to the class of the most similar labelled

examples. Characteristics of observations are collected for both training and test dataset'', adapted from *Zhang (2016)* In the second phase, the detected number plate is entered into the web application database constructed for the parking site. After entering the number plate information into the web application, the Internet of Things (IoT) sensors are activated, and the gate for the parking slots is opened.

These parking slots relate to motion sensors to identify whether a car is parked at the location. In the third phase, we use an Arduino circuit to identify an object's existence at a location. The display image screen at the parking gate entrance continuously monitors and displays information about the nearest available parking. The user can read the display board and move his car to the parking slot shown on the LED. The car's status is recorded until the sensors are activated and measures the availability of a vehicle at a parking location. Once the car has left and the sensor cannot identify any object in front of it, the Arduino routine updates the value of the nearest available parking in the zone area. Our model outperforms the state-of-the-art technique of ANPR.

## Data privacy and integrity

The data collected in this study refers to the hypothetical data created for testing purposes. The dataset taken from the open source data is used for the model's training. The testing is completed with a part of the dataset supported by the same dataset. The hypothetical data is considered to reserve an individual's privacy and data privacy. The independent dataset created for the testing of the model is stored in the form of a CSV file for classification and processing. The algorithm proposed in the study identifies the number plates using the ANPR algorithm and OCR techniques. The final system created is responsible for determining crowded parking locations. The number plate recognition is done in this case by considering several conditions:

- The position of the camera is one of the important considerations. This study used the camera location perpendicular to the arrival vehicle number plate.
- The camera's resolution is also one of the most important features in this study. We tested the model with a 1080 px camera mounted over the entrance location of the parking.
- The vehicle entering the parking location is assumed to be stationary during the algorithm's execution to analyze the number plate.
- The size of the number plate is considered to be the general size visible. The number plate should be clear, and no traces of dust or dirt are expected in this case.
- The area is expected to be lighted with bright, preferably sunlight. The testing is done in a zone with proper light to visualize the number. A part of the algorithm was also tested in sunlight, and the results were more accurate.
- The main module of the proposed study is the processing of the image captured by the camera. We tested the project with a system running on 16 GB RAM and an Intel Core i5 processor. Under ideal circumstances, the system could perform well with high accuracy and precision.

The security of the complete system information depends on the central server's database management system. The testing is done with a web application holding the information of the number plates using a simple website with MySQL database. The data is hosted by a secure web server protected by a firewall. Since this prototype is a miniature depiction of the central system, security is managed with a general firewall and database management system encryption system only. With the scalability and growth of the system, security will be enhanced using secure transaction certificates and advanced firewall software. The data stored in the centralized repository of number plates will be kept confidential to maintain the privacy of individual users.

### Future direction

For future work, several directions can be explored to enhance and expand the current system. First, extending the recognition system to support multi-script number plates, such as Chinese, Cyrillic, or Devanagari, would increase its global applicability. Additionally, optimizing the model for real-time recognition in challenging conditions like poor lighting, skewed angles, and motion blur, as well as integrating it with edge computing devices, could significantly enhance its usability in smart cities and intelligent transportation systems. Another potential direction involves the integration of the system with IoT networks for applications such as automated vehicle tracking, toll collection, and security monitoring. Incorporating advanced machine learning techniques, such as generative models or transformers, could improve the robustness of the system in complex scenarios like occlusion or weather-related issues. Additionally, developing language-specific optimizations and custom datasets for regions with unique number plate formats and layouts could further improve recognition accuracy. Finally, exploring zero-shot learning or transfer learning approaches may help address the challenge of data scarcity in certain languages or regions, allowing the system to generalize across different environments without the need for extensive retraining.

## CONCLUSION

The article presented in this study signifies and provides the complete details of implementing a machine learning-powered number plate recognition system. The prototype implementation and experiments yield results that prove that the model presented is of immense importance. The proposed model works into three phases: detection of the number plate, identification of the characters on the plate, and finally, recognition of these characters. After the recognition, the second phase consists of finding a proper location for the nearest available parking space inside the parking zone. Arduino and IoT-based sensors continuously monitor the most immediately available parking zone and space. Several experiments were conducted with variant weather conditions and light illumination locations. The system's performance is identified with the help of mathematical equations, and a comparison between the results is made to prove the model's accuracy. Compared to the existing model, the number of images processed in this model is almost three times the commercial values. However, the proposed model is in its early stages of development, so the system's scalability still needs to be achieved.

We look forward to scaling the system to a macro level, where the information relative to vehicles in a city will be integrated for analysis and parking. Thus, it proves that the study benefits commercial and business-oriented model replication. The comparison of the present model with the previously existing state-of-the-art solutions ensures that this model is more helpful for various sectors and commercial purposes. We anticipate that this research will contribute to advancing knowledge in computer vision and improve real-time object surveillance capabilities.

In future work, the model can be expanded to integrate additional sensors for real-time parking occupancy detection, enhancing its effectiveness in guiding drivers to available parking spaces. Moreover, exploring machine learning models for dynamic parking pricing could be valuable for optimizing parking utilization and revenue generation. Furthermore, scaling the model to predict and manage parking on a city-wide scale aligns with broader intelligent city initiatives, fostering more efficient urban mobility and reducing congestion. Additionally, conducting further experiments to ensure the robustness of the model at scale and addressing potential mismatches due to computer vision API issues will be crucial for its successful implementation in real-world scenarios. These advancements align with the objectives of Vision 2030 in Saudi Arabia and contribute to solving parking-related challenges.

### Funding
This work is supported by the Deanship of Scientific Research at Shaqra University. The funders had no role in study design, data collection and analysis, decision to publish, or preparation of the manuscript.

### Grant Disclosures
The following grant information was disclosed by the authors:
Deanship of Scientific Research at Shaqra University.

### Competing Interests
Khalid Raza is an Academic Editor for PeerJ Computer Science.

### Author Contributions

- Mofadal Alymani conceived and designed the experiments, performed the experiments, analyzed the data, performed the computation work, prepared figures and/or tables, authored or reviewed drafts of the article, and approved the final draft.
- Lenah Abdulaziz Almoqhem performed the experiments, performed the computation work, prepared figures and/or tables, and approved the final draft.
- Dhuha Ahmed Alabdulwahab performed the experiments, performed the computation work, authored or reviewed drafts of the article, and approved the final draft.
- Abdulrahman Abdullah Alghamdi conceived and designed the experiments, analyzed the data, authored or reviewed drafts of the article, and approved the final draft.

- Hussain Alshahrani analyzed the data, performed the computation work, authored or reviewed drafts of the article, and approved the final draft.
- Khalid Raza analyzed the data, prepared figures and/or tables, authored or reviewed drafts of the article, and approved the final draft.

## Data Availability

The data and code are available in the Supplementary Files.

## Supplemental Information

Supplemental information for this article can be found online at http://dx.doi.org/10.7717/peerj-cs.2544#supplemental-information.

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
