# Peer review of "Enabling smart parking for smart cities using Internet of Things (IoT) and machine learning"

_PeerJ Computer Science, doi:10.7717/peerj-cs.2544_

## Round 0.1 · original submission · Major Revisions

· Academic Editor

Major Revisions

Please follow review comments.

**Language Note:** The review process has identified that the English language must be improved. PeerJ can provide language editing services - please contact us at [email protected] for pricing (be sure to provide your manuscript number and title). Alternatively, you should make your own arrangements to improve the language quality and provide details in your response letter. – PeerJ Staff

Reviewer 1 ·

Basic reporting

The study presents an innovative approach to addressing the pressing issue of parking space scarcity in urban areas through the development of a machine learning-powered number plate recognition system. This system, designed to facilitate the entry and parking process, is a promising step towards smart parking solutions.

Overall, the paper needs improvements on several aspects, as explained below.

1. The study provides an overview of the system's functionality but lacks depth in the technical description of the Automatic Number Plate Recognition (ANPR) and Optical Character Recognition (OCR) algorithms. Detailed algorithmic workflows, parameter settings, and the rationale for specific choices would enhance the reader's understanding and the reproducibility of the study.
Including a schematic diagram of the system architecture would visually summarize the operational flow and component interactions, providing clarity on the system's design.

2. The study mentions the use of open-source data sources like Kaggle for training images but does not discuss data preprocessing, augmentation, or the distribution of the dataset (e.g., the number of plates per jurisdiction, lighting conditions). Addressing these aspects would strengthen the validity of the model's performance claims.
Discussion on data privacy and security measures, especially in handling and storing license plate information, is absent. Given the sensitivity of personal data, elaborating on these measures is critical for ethical and legal compliance.

3. The literature review of the study is very terse. Many studies have not discussed in this part. The authors should discuss the following studies: Video surveillance using deep transfer learning and deep domain adaptation: Internet of things enabled parking management system using long range wide area network for smart city; Optimizing the Identification and Utilization of Open Parking Spaces Through Advanced Machine Learning; Towards better generalization; Deep visual social distancing monitoring to combat COVID-19: A comprehensive survey; Detection of car parking space by using Hybrid Deep DenseNet Optimization algorithm

Experimental design

4. While the study reports high classification accuracy rates and improved processing times, a comparative analysis with baseline models is missing. Providing performance metrics such as precision, recall, and F1-score alongside accuracy, and comparing these with state-of-the-art systems, would offer a more comprehensive evaluation.
The impact of varying environmental conditions on system performance is mentioned; however, quantitative results showcasing the system's robustness under these conditions are lacking. Including these results would validate the system's operational effectiveness in real-world scenarios.

Validity of the findings

5. The application of the system in real-world settings such as government buildings, malls, and airports is discussed, yet there is no mention of pilot testing or real-world deployment feedback. Incorporating case studies or pilot project results would substantiate the system's practical viability and user acceptance.
Scalability discussions are limited. As the system is proposed for deployment in densely populated areas, considerations for scaling, including managing large volumes of concurrent requests and data, should be addressed.


6. The conclusion highlights the intent to adapt the system for Arabian number plates and conduct further experiments for model robustness. Expanding on specific future work directions, such as integrating additional sensors for real-time parking occupancy detection or exploring machine learning models for dynamic parking pricing, could inspire subsequent research.

Reviewer 2 ·

Basic reporting

The authors treated the problem of the parking management system. This system recognizes the vehicles that can be entered into parking through its plate. A simple Arduino application is described in this paper.

1) Figure 2: delete this figure, the ERR system is simple, and no need to represent it by a figure.
2) Figure 4: delete this figure, the AD system is simple, and no need to represent it by a figure.
3) The proposed algorithm is simple.
4) A simple application of Arduino can’t be a research work.
5) The literature study is not sufficient to compare between the previous work and the proposed one.
6) I can’t see the contribution of the paper once the studied problem is already known and treated in several parking in the world.
7) Without comparison with other studies the research work can’t be assessed correctly.
8) The work is a simple application of trivial code using Arduino.

Experimental design

The experimental designs are not sufficient to be a research work

Validity of the findings

No novelty in the studied problem.
The studied problem is well known and used in several parking in the world. In addition, the paper doesn't present any novel algorithms.

Additional comments

I recommend to reject the paper

---

## Round 0.2 · Major Revisions

· Academic Editor

Major Revisions

Please follow reviewer requests.

Reviewer 1 has suggested that you cite specific references. You are welcome to add it/them if you believe they are relevant. However, you are not required to include these citations, and if you do not include them, this will not influence my decision.

**Language Note:** The review process has identified that the English language must be improved. PeerJ can provide language editing services - please contact us at [email protected] for pricing (be sure to provide your manuscript number and title). Alternatively, you should make your own arrangements to improve the language quality and provide details in your response letter. – PeerJ Staff

Reviewer 1 ·

Basic reporting

The study introduces a machine learning-powered system to address the issue of parking space scarcity in urban areas by implementing an Automatic Number Plate Recognition (ANPR) system and a real-time parking space availability display. This two-part solution aims to streamline vehicle entry and minimize the time drivers spend searching for parking. The system utilizes cameras for license plate recognition via optical character recognition (OCR) and vacant space detection, storing this data in a centralized database to display available spots at the parking entrance. The experimental evaluation showed high accuracy in license plate recognition, with performance comparisons between SVM and YOLOv8 models.

Experimental design

1. The literature review of this article is very terse. This section should be expanded by discussing the following studies:

Smart speed camera based on automatic number plate recognition for residential compounds and institutions inside Qatar; Video surveillance using deep transfer learning and deep domain adaptation: Towards better generalization; Deep visual social distancing monitoring to combat COVID-19: A comprehensive survey; An IoT assisted intelligent parking system (IPS) for smart cities; Smart cities: The role of Internet of Things and machine learning in realizing a data-centric smart environment; A trust-based secure parking allocation for IoT-enabled sustainable smart cities; Deep reinforcement learning approach towards a smart parking architecture; Deep reinforcement learning-based long-range autonomous valet parking for smart cities

2. The study mentions the system is in early development stages, with scalability not yet achieved. Testing on a small scale might not accurately reflect real-world complexities and variability in parking situations.

Validity of the findings

3. While open datasets offer diversity, the reliance on a dataset of 355 images for testing might not sufficiently cover the vast range of real-world scenarios, including varying license plate designs and conditions.

4. The system's performance under ideal conditions (good lighting, stationary vehicles, clean number plates) raises questions about its efficacy in less-than-ideal scenarios, such as poor weather, moving vehicles, or dirty plates.

5. The system's reliance on specific hardware (Intel i5, 16GB RAM) for optimal performance could limit its deployment in environments with lower technical specifications.

6. While the study touches on data privacy and system security, the depth of these considerations seems insufficient, especially considering the sensitivity of tracking and storing vehicle movements and license plate data.

---

## Round 0.3 · Major Revisions

· Academic Editor

Major Revisions

The recommendations of reviewer 2 must be addressed before publication.

Reviewer 1 ·

Basic reporting

The paper has significantly been improve after addressing the reviewers' comments. I thank the authors for their efforts.

Experimental design

This part is very comprehensive and analyzes the performance of the proposed method under different circumstances; no further tests are required.

Validity of the findings

The presented results showcase the performance of the proposed method and confirm the hypothesis presented in the proposed method section.

Reviewer 2 ·

Basic reporting

The authors treated the problem of the parking management system. This system recognizes the vehicles that can be entered into parking through its plate. A simple Arduino application is described in this paper.

1) Figure 2: delete this figure, the ERR system is simple, and no need to represent it by a figure.
2) Figure 4: delete this figure, the AD system is simple, and no need to represent it by a figure.
3) The proposed algorithm is simple.
4) A simple application of Arduino can’t be a research work.
5) The literature study is not sufficient to compare between the previous work and the proposed one.
6) I can’t see the contribution of the paper once the studied problem is already known and treated in several parking in the world.
7) Without comparison with other studies the research work can’t be assessed correctly.
8) The work is a simple application of trivial code using Arduino.

Experimental design

The experimental designs are not sufficient to be a research work

Validity of the findings

No novelty in the studied problem.
The studied problem is well known and used in several parking in the world. In addition, the paper doesn't present any novel algorithms.

Additional comments

I recommend to reject the paper

---

## Round 0.4 · Major Revisions

· Academic Editor

Major Revisions

This is the author's last chance to follow the review comments

Reviewer 2 ·

Basic reporting

Dear authors,

Unfortunately, I don't receive point-by-point responses to my 8 queries. I receive only 3 responses among 8. In addition, these 3 responses are general and not studied in scientific depth. I asked about a scientific comparison between your work and the published work.


Regards

Experimental design

no comment

Validity of the findings

no comment

Reviewer 3 ·

Basic reporting

The research integrates multiple technologies, including Python, OpenCV, TensorFlow, Easy OCR, SQLite, and Google Colab, to develop a comprehensive system for automated parking management. The system includes the development of an Automatic Number Plate Recognition (ANPR) system using OCR for seamless entry authentication and the implementation of a system to detect and display vacant parking spaces in real-time using cameras and a centralized database. The proposed system aims to reduce the time and effort spent by users in finding parking spaces, thus enhancing overall urban mobility and convenience. Additionally, the system ensures that only authorized vehicles can enter the parking lot, enhancing security and reducing the need for manual verification. However, several critical issues need to be addressed:
1. Research Contribution and Innovation: The paper lacks a clear statement of the research contribution and innovation. It appears to focus on the implementation of existing machine learning algorithms on available datasets without highlighting the unique aspects or advancements introduced by this research.
2. State-of-the-Art Review: The paper lacks a comprehensive review of state-of-the-art (2023-2024) solutions for the specific problem addressed. Including a review of relevant literature and comparing the proposed solution with existing ones would provide a better context for the research contributions.

I. Line 142 Missing Citation: It appears that there is a reference or citation missing at line 142. Please ensure that all claims or statements are properly supported with appropriate citations before finalizing the document.
II. Problem in Citation at Line 428: There seems to be an issue with the citation formatting or writing at line 428. Please review and correct any errors in citation style or formatting before submitting the document.

Experimental design

3. Dataset Properties: The author mentions using two datasets for training (lines 272 and 275), specifically from Kaggle and Roboflow, but does not provide details on the properties of these datasets, such as the number of samples and preprocessing steps applied.

4. System Testing and Validation: There is no discussion on the testing and validation of the proposed system. The paper should include scenarios where the system succeeds and fails, along with suggested solutions for cases where the system fails to correctly identify the car's license plate.
5. Citations and References: There is a notable lack of citations, particularly for the algorithms used, such as the K-Nearest Neighbor (KNN) model mentioned in line 268 and the YOLOv8 algorithm. Providing proper citations and background information on these algorithms is necessary for clarity and credibility. Considering:

Validity of the findings

6. Comparison with Existing Solutions: The author should include a comparison between their proposed solution and existing solutions. This comparison would help to highlight the advantages and potential improvements offered by the proposed system.
7. Dataset for Training and Testing: Clarification is needed on the datasets used for training and testing. If the datasets were collected by the authors, details about the collection process should be provided. If not, the paper should explain how the authors plan to achieve one of their research goals by applying this system in Saudi Arabia (KSA), considering that vehicle license plates in KSA contain both Arabic and English characters. The paper should discuss the proposed solution for dealing with these multilingual plates as part of their goal aligned with KSA's Vision 2030.
8. Author mentioned, their research involves using IoT; The system’s performance was evaluated using mathematical equations, and comparisons were made to prove the model’s accuracy (there is no comparison with the existing methods). They mentioned, compared to existing models, this model processes nearly three times the number of images as commercial systems. However, the only implementations mentioned are the Arduino and IoT-based sensors for continuously monitoring the most immediately available parking zones and spaces (there is no details). They mentioned also, several experiments were conducted under varying weather conditions and light illumination scenarios without details.

Additional comments

To further enhance their work, the author should review the following papers:
1. "Improving parking availability prediction in smart cities with IoT and ensemble-based model"
2. "Revolutionizing Urban Mobility: IoT-Enhanced Autonomous Parking Solutions with Transfer Learning for Smart Cities"
3. "Optimizing the Identification and Utilization of Open Parking Spaces Through Advanced Machine Learning"
4. "Smart cities: The role of Internet of Things and machine learning in realizing a data-centric smart environment"
5. "Internet of things enabled parking management system using long range wide area network for smart city"
6. These references could provide valuable insights and methodologies to improve the integration and application of IoT in their research.

Annotated reviews are not available for download in order to protect the identity of reviewers who chose to remain anonymous.

---

## Round 0.5 · Major Revisions

· Academic Editor

Major Revisions

Please address the remaining issues within the manuscript.

Reviewer 3 ·

Basic reporting

Thank you for sharing your work on addressing the pressing issue of parking congestion in urban areas. Your proposed system, which utilizes Automatic Number Plate Recognition (ANPR) and real-time parking space detection, presents an innovative approach to enhancing urban mobility. The work is appreciated; However, there are several areas where improvements are needed for the paper to meet the standards of a high-quality research journal such as PeerJ Computer Science. I would like to recommend that you further develop your research paper to clearly articulate the technical soundness, novelty, and contributions of your work. Aspects such as the specific deep learning models used, the comparison with recent state-of-the-art frameworks, and the detailed explanation of your methodology’s advantages over existing solutions are currently not sufficiently addressed. Furthermore, I suggest that you address the reviewer comments more thoroughly in your revisions. By providing detailed responses and incorporating the feedback, you will be able to enhance the clarity, depth, and overall quality of your paper, making it more suitable for publication in a reputable journal. These points will strengthen the paper’s impact and demonstrate its contribution to the field of smart parking systems.

Experimental design

The methodology section of the manuscript requires a significant amount of additional detail. Specifically, the authors need to provide comprehensive information on the following aspects:
1. Dataset Description: Provide a detailed description of the dataset, including its composition, size, and structure. It is important to include a visual representation (such as a figure) to help readers understand the dataset better.
2. Preprocessing Steps: Clearly outline the preprocessing techniques used to prepare the dataset. Include specifics on any data cleaning, normalization, or augmentation methods applied, along with visual examples if possible.
3. Feature Extraction: Describe the feature extraction process in detail, including the number of features extracted. Provide any relevant figures or tables that illustrate the features and their significance.
4. Feature Selection: Explain the feature selection process used to identify the most relevant features. Discuss why certain features were selected and how this selection impacts the overall performance of the system.
The current version of the manuscript lacks sufficient detail in these areas. For a more comprehensive approach, I recommend reviewing the following work for additional insights: "Intelligent System for Vehicles Number Plate Detection and Recognition Using Convolutional Neural Networks" by Nur-A-Alam, Mominul Ahsan, Md. Abdul Based, and Julfikar Haider.
A low cost IoT-based Arabic license plate recognition model for smart parking systems “A low cost IoT-based Arabic license plate recognition model for smart parking systems - ScienceDirect” https://doi.org/10.1016/j.asej.2023.102178

This reference may provide valuable guidance on how to effectively present these details in the methodology section.
To strengthen the manuscript, the authors should focus on improving the methodology section. Clearer presentation of the methodology will help in demonstrating the contribution and novelty of the work. A detailed and well-justified approach will be crucial for the paper to be considered for publication.

Validity of the findings

The current results are not sufficient for publication in a reputable journal. I recommend the author include additional results and compare their findings with recent frameworks based on advanced deep learning models such as EfficientDet, and Transformer-based approaches like DETR. For example (time : The time taken for image capture, plate recognition, and database lookup may introduce delays. Optimizing the system for faster processing times is necessary to enhance user experience and operational efficiency.) Moreover, incorporating techniques for vehicle and license plate detection used in smart parking systems, such as SSD and Faster R-CNN, would provide a more comprehensive evaluation and demonstrate the effectiveness of their proposed method. Additionally, it would be beneficial to include examples demonstrating both successful detections and failure scenarios of the system. For example : the accuracy of the ANPR and OCR systems can be affected by various factors such as lighting conditions, plate cleanliness, and font styles. These limitations might lead to false negatives or false positives, affecting the reliability of access control.
I also recommend that you consider incorporating Arabic numbers in your proposed method. This enhancement would reflect positively on the contribution and novelty of your model, making it more suitable for regions such as Saudi Arabia, which aims to achieve the Saudi Vision 2030. Given that vehicle plates in Saudi Arabia include Arabic numbers, this adaptation would ensure better applicability and relevance of your system in that context."
Improved Comparisons:
The paper would benefit from a more thorough comparison with related work. Ensure that comparisons are conducted using similar datasets or by applying the proposed model to datasets from other studies. Presenting and discussing the results from these comparisons will provide a clearer context for evaluating the effectiveness and relevance of the proposed method.
As part of improving the manuscript, I recommend that the authors review the following article for comparison and additional insights:
(IJACSA) International Journal of Advanced Computer Science and Applications, Vol. 14, No. 4, 2023, pp. 8 An Automatic Framework for Number Plate Detection using OCR and Deep Learning Approach Available at: www.ijacsa.thesai.org
I recommend the author review the following article: "Edge-Based License-Plate Template Matching for Identifying Similar Vehicles." The article can be accessed at https://doi.org/10.3390/vehicles3040039.
These articles presents a comprehensive framework for number plate detection using OCR and deep learning techniques. Comparing your methodology and results with those presented in this article will help strengthen your manuscript by providing a benchmark and potentially identifying areas for improvement.

Significant enhancements are needed in the methodology, novelty, and comparative analysis sections of the manuscript. I encourage the authors to address these issues thoroughly to improve the paper's suitability for publication in a reputable journal.

Additional comments

It seems that many of the reviewers' comments have not been adequately addressed. I strongly encourage the authors to carefully review and address all feedback provided by the reviewers. As mentioned in the previous comments, the authors should present case studies or examples illustrating scenarios where the system successfully classifies plates and where it might fail. This explanation is crucial for understanding the system's limitations and strengths. Unfortunately, this comment has not been addressed or presented in the current version of the article. This will significantly enhance the quality and credibility of the manuscript.

Research Depth and Novelty:
The current manuscript seems to focus more on application rather than presenting substantial research contributions. The novelty and contribution of the work are not clearly articulated. It is essential to highlight how this work advances the field or offers new insights that are not covered by existing literature.

Algorithm and Comparison:
The algorithm employed appears to be relatively simple, and the methodology for comparison is also quite basic. Recent advancements in this area may provide more sophisticated approaches that could be considered. Additionally, the comparison with other works is not sufficiently robust. A more rigorous comparison should be performed, ideally using the same dataset or applying the proposed model to datasets from other studies to validate its effectiveness.

Annotated reviews are not available for download in order to protect the identity of reviewers who chose to remain anonymous.

---

## Round 0.6 · Major Revisions

· Academic Editor

Major Revisions

Kindly do your best to address the manuscript as per the reviewer 3 suggestion and resubmit it.

Reviewer 3 ·

Basic reporting

The manuscript presents a proposed solution for enabling smart parking systems in urban environments through the integration of IoT and machine learning technologies. As cities face increasing vehicle numbers and limited parking spaces, the need for efficient parking management has become critical. This work aims to address the challenges of parking congestion by utilizing Automatic Number Plate Recognition (ANPR) and real-time monitoring of parking availability.

However, upon review, it is evident that the manuscript does not yet meet the standards required for publication in a reputable journal.

Experimental design

1. Dataset Quality: The uploaded dataset contains very few samples, which is insufficient for robust analysis. More comprehensive data is necessary for meaningful results.
2. Code Availability: The code provided lacks updates that accommodate both Arabic and English number recognition. Full access to the complete and updated code is essential.
3. Clarity of Novelty and Contribution: The manuscript does not clearly articulate its novelty and contributions. The work appears more as an application rather than a research study. Enhancements in methodology and clarity of purpose are recommended.
4. Code and Dataset Submission: Authors should upload the complete code and the dataset used for training and testing to ensure reproducibility and validation of the results.
5. Results and Analysis: The results and analysis presented for verification are insufficient for publication. We recommend revisiting the recent paper titled "Automatic Number Plate Detection and Recognition Using YOLO World" for guidance on improving the analysis:
6. Agarwal, V., & Bansal, G. (2024). Automatic number plate detection and recognition using YOLO world. Computers and Electrical Engineering, 120, 109646. Elsevier.

Validity of the findings

7. Comparison with Other Datasets: In the comparison section, authors should test their proposed model against other datasets and provide a thorough comparison of results.
8. Response to Feedback: We hope the authors consider these comments seriously and make clear modifications based on this feedback. Evidence of these changes in the methodology, code, and data should be provided.
9. Enhancement of Results and Analysis: Further enhancement of the results and analysis is necessary. Please refer to the aforementioned paper for insights on improving this aspect.

Additional comments

The reviewer has given the authors more than three opportunities for revision, yet the manuscript remains at the same level of inadequacy. While some enhancements have been made, it is still not sufficient for publication in a reputable journal. We recommend that the authors revise their manuscript to align with the quality of work typically published in high-impact journals. It is crucial for the authors to focus on strengthening their methodology and clearly articulating their contributions and novelty.

Annotated reviews are not available for download in order to protect the identity of reviewers who chose to remain anonymous.

---

## Round 0.7 · accepted · Accept

· Academic Editor

Accept

Author has addressed reviewer comments properly. Thus I recommend publication of the manuscript.